health and disease and epidemiology/evolution/ecology

*Lepeophtheirus salmonis*, multiresistance, salmonid aquaculture, organophosphate, pyrethroid

**Author for correspondence:**
Helene Børretzen Fjørtoft
e-mail: helene.b.fjortoft@ntnu.no

# Losing the 'arms race': multiresistant salmon lice are dispersed throughout the North Atlantic Ocean

Helene Børretzen Fjørtoft[1,2], Frank Nilsen[2], Francois Besnier[3], Anne Stene[1], Ann-Kristin Tveten[1], Pål Arne Bjørn[3], Vidar Teis Aspehaug[4] and Kevin Alan Glover[2,3]

[1]Department of Biological Sciences in Aalesund, Norwegian University of Science and Technology, PO Box 1517, 6025 Aalesund, Norway
[2]Department of Biology, Sea Lice Research Center, University of Bergen, PO Box 7803, 5020 Bergen, Norway
[3]Institute of Marine Research, PO Box 1870, 5817 Bergen, Norway
[4]PatoGen AS, PO Box 548, 6001 Aalesund, Norway

HBF, 0000-0001-8195-0582

Nothing lasts forever, including the effect of chemicals aimed to control pests in food production. As old pesticides have been compromised by emerging resistance, new ones have been introduced and turned the odds back in our favour. With time, however, some pests have developed multi-pesticide resistance, challenging our ability to control them. In salmonid aquaculture, the ectoparasitic salmon louse has developed resistance to most of the available delousing compounds. The discovery of genetic markers associated with resistance to organophosphates and pyrethroids made it possible for us to investigate simultaneous resistance to both compounds in approximately 2000 samples of salmon lice from throughout the North Atlantic in the years 2000–2016. We observed widespread and increasing multiresistance on the European side of the Atlantic, particularly in areas with intensive aquaculture. Multiresistant lice were also found on wild Atlantic salmon and sea trout, and also on farmed salmonid hosts in areas where delousing chemicals have not been used. In areas with intensive aquaculture, there are almost no lice left that are sensitive to both compounds. These results demonstrate the speed to which this parasite can develop widespread multiresistance, illustrating why the aquaculture industry has repeatedly lost the arms race with this highly problematic parasite.

# 1. Introduction

Human health and food security have long been challenged by arthropods. Insects and arachnoids destroy crops and act as vectors of diseases in humans and livestock alike. The use of pesticides has helped reduce negative impacts, but nevertheless, an estimated 18–20% of the world's crop production is lost to insects and mites [1], and the WHO estimates that 700 000 human lives are lost annually due to vector-borne diseases [2].

Pesticides that are used over time will lose their effect as target organisms develop reduced sensitivity and ultimately resistance. As established chemotherapeutants become less effective, new pesticides are developed and implemented, which help controlling pest populations for a limited time. However, ultimately the new chemicals lose their efficiency, and as a result, the pest eventually becomes multiresistant [3]. An additional obstacle to the development of new pesticides is cross-resistance [4], where a chemical may be ineffective if similar chemical groups (or with similar mode of action) have already been used. The development of new and novel pesticides is both costly and time consuming, and all chemicals are not available globally [5]. As a result, the development of multiresistant pest populations leaves farmers and health workers with few tools to control important pest or vector populations [6].

Arthropods are also a pest in fish farming. In salmonid aquaculture, the Caligids *Lepeophtheirus salmonis*, *Caligus elongatus* and *C. rogercresseyi* represent persistent parasites on both farmed fish and nearby wild populations [7,8]. *Lepeophtheirus salmonis* (hereon referred to as the salmon louse) is a copepod with a natural distribution throughout the North Atlantic, with its closely related subspecies in the North Pacific [9]. It is a specialist parasite of salmonids, which in the North Atlantic includes Atlantic salmon (*Salmo salar*), sea trout (*Salmo trutta*), Arctic charr (*Salvelinus alpinus*) and non-native farmed rainbow trout (*Oncorhynchus mykiss*). The life cycle of the salmon louse consists of eight stages [10]. It begins with three planktonic stages, where the third, the copepodite, is the infective stage. If the copepodite finds a suitable host, it will attach itself and moult through two sessile stages. The louse becomes mobile after reaching the first pre-adult stage, meaning that it can move around on the host. After the second pre-adult stage, the louse goes through a final moult into the adult stage. The adult female produces egg strings that are attached to her abdomen. Each set of egg strings may contain 300–600 eggs [11]. The salmon louse thus has two modes of dispersal: first during the planktonic stages, where it will follow the current until it localizes a host, and second as a hitchhiker on the host fish while reproducing. Farmed salmonids remain in their pens, but wild sea trout and charr roam through the fjords and coastal areas and the wild Atlantic salmon will transport the salmon louse to the high seas, where salmon from all parts of the North Atlantic meet.

Before the onset of aquaculture, wild Atlantic salmon, sea trout and Arctic charr were largely available as hosts for salmon lice during the spring, summer and autumn months. The winters represented bottlenecks for lice population development as most wild salmonids were either in the ocean or in fresh water. However, with the introduction of Atlantic salmon and rainbow trout farming in coastal open net-pens throughout the year, both the number and seasonal availability of hosts increased, and salmon lice epizootics were quickly reported on farms. Pesticides first used in agriculture were adapted and applied in aquaculture with good effect [12]. The first compounds were based on organophosphates. However, like in agriculture, a reduction in effect was noted after a relatively short time [13]. New delousing chemicals were thereafter introduced to replace organophosphates; first pyrethroids, later hydrogen peroxide and finally emamectin benzoate and flubenzurones in 1999 [14]. However, resistance has subsequently been documented for pyrethroids, emamectin benzoate and hydrogen peroxide [15–18]. As novel compounds were introduced, the use of the old almost stopped, shifting the selection for resistance to the new chemical. For pyrethroids, reduced sensitivity was registered already 6 years after the first commercial use, and for emamectin benzoate, resistance was dispersed throughout the North Atlantic in less than 11 years after the introduction [14,19,20]. From 2000 onwards, emamectin benzoate was the preferred compound, but when problems with resistance arose, there were no new compounds available. Instead, organophosphates, pyrethroids and hydrogen peroxide were reintroduced. The initial effect was good, but the problems with resistance reappeared quickly [21].

Recently, genetic markers associated with resistance to organophosphates and pyrethroids have been discovered and validated in the salmon louse [22,23]. The development of TaqMan assays to screen for these mutations allows a large number of individuals to be tested effectively and provides a novel tool to map the distribution and prevalence of resistance to the relevant compounds both in time and space. The distribution and prevalence of the mutations associated with resistance to organophosphates and

pyrethroids have earlier been analysed separately [24–27]. However, in agriculture, it is not unusual that some pest species develop resistance to multiple compounds [28,29]. Resistance and multiresistance is selected for by intensive use of the relevant chemicals. Tables 1 and 2 give an overview of the use of organophosphates and pyrethroids in salmonid aquaculture throughout the North Atlantic from the introduction of emamectin benzoate. However, the available information and level of detail regarding the use of delousing chemicals varied greatly among the salmon producing countries.

Multiresistant salmon lice were described in controlled laboratory experiments where a laboratory strain resistant to pyrethroids was crossed with a laboratory strain resistant to organophosphates [30]. Multiresistance in salmon lice has also been suspected in the field, based on the presence of reduced sensitivity towards multiple compounds in salmon lice within the same region [31]. However, thus far, this has not been documented. Given the fact that this parasite represents the largest single challenge to further Atlantic salmon aquaculture production [32], and, the largest single challenge to environmentally sustainable salmonid aquaculture [33,34], investigating the prevalence and distribution of multiresistance in lice throughout the North Atlantic represents a timely endeavour. In order to address this, data from approximately 2000 salmon lice sampled from wild Atlantic salmon, sea trout and farmed Atlantic salmon from across the North Atlantic was examined for simultaneous appearance of genetic markers associated with resistance to organophosphates and pyrethroids.

# 2. Material and methods

## 2.1. Samples

A total of 1988 salmon lice sampled throughout the North Atlantic in the period 2000–2017 were analysed with the genetic markers for both pyrethroid and organophosphate resistance. A full overview of samples is provided (electronic supplementary material, table S1).

Lice were sampled from fish farms in Ireland, Scotland, the Faroes and Iceland in 2016, and from Atlantic Canada in January 2017. Samples were taken during routine operations or by fish vet personnel. A sample from wild Atlantic salmon from the west coast of Greenland in 2016 was also obtained. The Greenlandic salmon lice were sampled by researchers from NOAA from wild salmon caught by local fishermen [24]. From each of these countries, 66–69 lice were analysed, and the whole North Atlantic sample set from the years 2016 and 2017 numbered 399 lice.

From Norway, a total of 532 lice from wild sea trout and 304 lice from returning wild Atlantic salmon sampled in 2014 from nine regions along the Norwegian coast were analysed. All host fish were captured as part of ongoing research programmes or licensed fisheries [24]. A maximum of 10 lice were sampled from each host, and a scale sample was taken to determine sea age and reveal escaped farmed salmon [35]. Salmon lice from both host species were not available from all regions. The numbers ranged from 12 to 50 lice per region from Atlantic salmon, and from 48 to 185 for lice from sea trout.

A historical sample set of 753 lice sampled from Atlantic Canada, Ireland, Shetland, Scotland, the Faroes, Norway and Russia was also genotyped. These lice were sampled between the years 2000 and 2009 and have previously been used in population genetics and genomics studies [20,36,37]. All samples were obtained from fish farms with the consent of the owner, except the sample from Russia. This sample was collected from returning wild Atlantic salmon caught and killed by local fishermen [36]. From each country or region, 29–96 lice were analysed.

## 2.2. Genetic analysis

All the 1988 lice samples described above were genotyped by the accredited test laboratory of PatoGen AS using their patented assays for pyrethroid and organophosphate resistance. The TaqMan assay for resistance to pyrethroids tests for the silent mutation *C14065T* in the mitochondrial *cytB* gene, which although not causative, is tightly linked with resistance to pyrethroids [22]. The TaqMan assay for organophosphate tests for the mutation *Phe362Tyr* in the *ace1a* gene, which is the cause of resistance to organophosphates [22].

Genotyping was performed as a one-step amplification (45 cycles) on an Applied Biosystems 7500 Real-Time PCR System. The pyrethroid resistance test used forward primer: 5′-TTC TTA CAG ACA AAG CTA AAG CCA CTA-3′, reverse primer: 5′-AGT AAC TCC TGC TCA CAT TCA ACC T-3′ and probe: 5′-CCC CCC C/T AAC TTA T-3′ [23]. Lice were classified as either resistant or sensitive. The organophosphate resistance test used forward primer: 5′-ATT TTA ATT GGA GCG AAT AAG GAA-3′, reverse primer: 5′-CGC TCT CCG TAT TTT TAA AGA GAT CT-3′ and probe: 5′-AAG GGA ATT

**Table 1.** Organophosphate use in North Atlantic salmonid aquaculture (kg active ingredient). NA, not available information. Sources are listed in the electronic supplementary material.

| | 2000 | 2001 | 2002 | 2003 | 2004 | 2005 | 2006 | 2007 | 2008 | 2009 | 2010 | 2011 | 2012 | 2013 | 2014 | 2015 | 2016 |
|---|---|---|---|---|---|---|---|---|---|---|---|---|---|---|---|---|---|
| Canada | 0 | 0 | 0 | 0 | 0 | 0 | 0 | 0 | 0 | NA | NA | NA | NA | NA | NA | NA | 858.7 |
| Faroe Islands | 0 | 0 | 0 | 0 | 0 | 0 | 0 | 0 | 0 | 14.04 | 23.25 | 113.25 | 55.5 | 11 | 209.5 | 443.4 | 408 |
| Iceland | 0 | 0 | 0 | 0 | 0 | 0 | 0 | 0 | 0 | 0 | 0 | 0 | 0 | 0 | 0 | 0 | 0 |
| Ireland | 0 | 0 | 0 | 0 | 0 | 0 | 0 | 0 | NA | NA | NA | NA | NA | 0 | 0 | 0 | 0 |
| Scotland | NA | NA | 0 | 0 | 0 | 0 | 0 | 0 | 34.25 | 155.22 | 91.95 | 125.95 | 166.83 | 121.41 | 225.55 | 205.19 | 391.46 |
| Shetland | NA | NA | 0 | 0 | 0 | 0 | 0 | 0 | 65.95 | 47.88 | 73.35 | 94.25 | 25.07 | 32.09 | 28.45 | 79.71 | 46.94 |
| Norway | 0 | 0 | 0 | 0 | 0 | 0 | 0 | 0 | 66 | 1884 | 3346 | 2437 | 4059 | 3037 | 4630 | 3904 | 1269 |

**Table 2.** Pyrethroid use in North Atlantic salmonid aquaculture (kg active ingredient). NA, not available information. Sources are listed in the electronic supplementary material.

| | 2000 | 2001 | 2002 | 2003 | 2004 | 2005 | 2006 | 2007 | 2008 | 2009 | 2010 | 2011 | 2012 | 2013 | 2014 | 2015 | 2016 |
|---|---|---|---|---|---|---|---|---|---|---|---|---|---|---|---|---|---|
| Canada | 0 | 0 | 0 | 0 | 0 | 0 | 0 | 0 | 0 | 0 | NA | NA | 0 | 0 | 0 | 0 | 0 |
| Faroe Islands | 8.7 | 11.7 | 17.9 | 15.3 | 10.5 | 1.07 | 2.36 | 4.56 | 4.92 | 4.63 | 24.25 | 27.97 | 0.08 | 14 | 7.69 | 15.27 | 12.59 |
| Iceland | 0 | 0 | 0 | 0 | 0 | 0 | 0 | 0 | 0 | 0 | 0 | 0 | 0 | 0 | 0 | 0 | 0 |
| Ireland | NA | NA | NA | NA | 1.83 | 2.44 | 3.3 | NA | NA | NA | NA | NA | NA | NA | NA | NA | NA |
| Scotland | | | 0 | 0 | 0 | 0.43 | 0.69 | 28.2 | 20.54 | 23.6 | 23.71 | 21.16 | 20.18 | 12.58 | 17.48 | 12.41 | 7.66 |
| Shetland | | | 0 | 0 | 0 | 0 | 0 | 9.2 | 0.826 | 1.467 | 2.42 | 1.32 | 0.87 | 0.132 | 0.003 | 0.007 | 0 |
| Norway | NA | 98 | 85 | 75 | 72 | 61 | 72 | 59 | 71 | 150 | 168 | 102 | 353 | 347 | 320 | 200 | 91 |

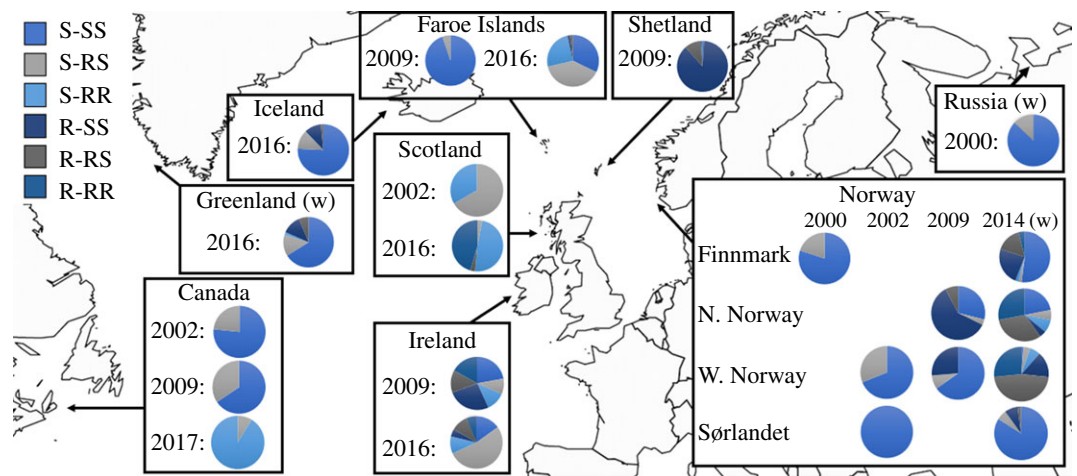

**Figure 1.** The frequency of salmon lice carrying genotypes associated with organophosphate resistance (RS or RR), pyrethroid resistance (R), both = multiresistant (R-RR or R-RS) or none = sensitive (S and SS). The map demonstrates the dispersion of resistance and multiresistance both in time and space in the North Atlantic from 2000 to 2017. Samples marked with (w) are from wild salmonids; wild Atlantic salmon for Greenland and Russia, and wild sea trout for Norway 2014.

ATT T/A CAT CAT G-3′ [38]. The test labelled each individual louse as homozygote sensitive (SS), heterozygote partly resistant (RS) or homozygote resistant (RR) to organophosphates. Here, we classified a salmon louse as multiresistant when it was defined as resistant in the pyrethroid test and carrying the RS or RR genotype for organophosphates.

## 2.3. Statistics

Multiresistance was considered as a binary variable where genotypes with at least one R allele at each locus (R-RR and R-RS) were considered multiresistant, all other genotypes were non-multiresistant. The binary multiresistance was then modelled as a response to sampling year and site, in a general linear model (GLM) with binomial family. One GLM was fitted for all data from the North Atlantic and another for the data from Norway. Within Norway, the effect of sampling site was only tested within the 2014 samples as this was the only sampling year with sufficient geographical variation.

# 3. Results

## 3.1. Contemporary samples

In the samples from 2016 to 2017, which is the most recent material from the North Atlantic, multiresistant lice were found on salmonids in all countries except Canada (figure 1). However, in both the Icelandic and Faroese samples, only one multiresistant louse was found. The highest frequency of multiresistant lice was found in Scotland (48%). No chemical-sensitive lice were observed in Scotland and Canada, as all the tested lice were resistant to organophosphates. Iceland, on the other hand, had a frequency of 88% sensitive lice.

In the salmon lice sampled from wild salmonids in Norway in 2014, multiresistance was found in all regions tested, both in lice sampled from sea trout and Atlantic salmon (figure 2). However, the frequency varied greatly among regions and host types. For example, only two multiresistant lice were observed in the samples from Sørlandet, one from sea trout (2% of the sample) and one from salmon (10%), whereas 74% of the lice from sea trout in Trondheimsfjord were multiresistant. In general, the frequencies of multiresistant lice on wild salmonids were greater in aquaculture-intensive regions (electronic supplementary material, figure S1), and the frequencies were in general higher on wild sea trout than wild Atlantic salmon. In Norway, the lice sampled from sea trout at Sørlandet, a region with almost no aquaculture, had the highest frequency of lice sensitive to both compounds (84%).

## 3.2. Historical samples

In the historical North Atlantic material, resistance to organophosphates was detected already in the samples from 2000, but pyrethroid resistance was not detected until the 2009 samples. The first

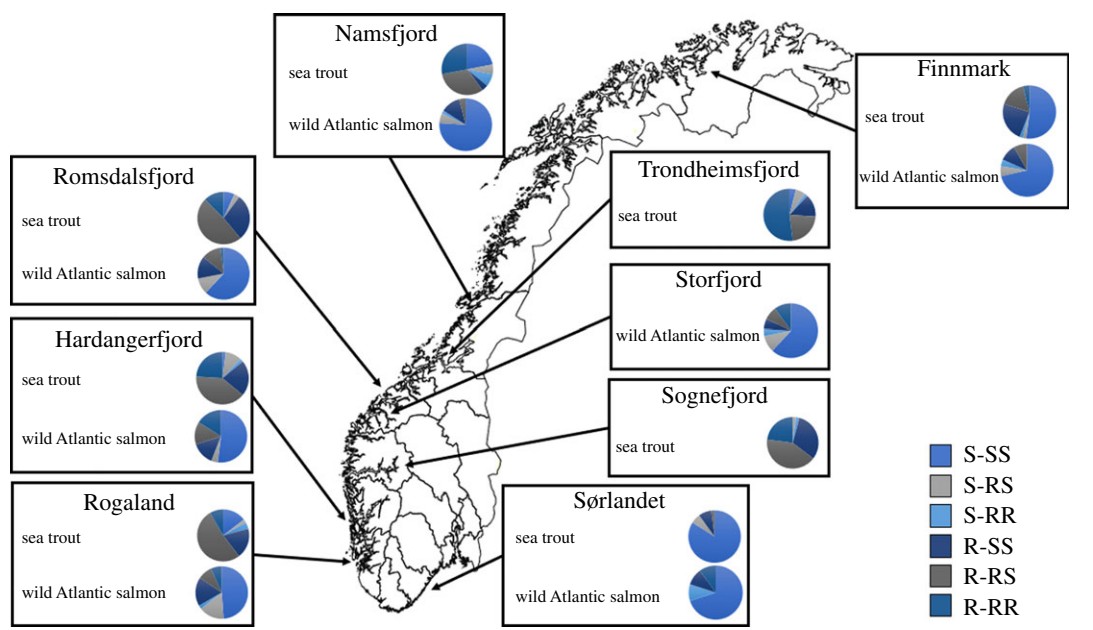

**Figure 2.** The prevalence of genotypes associated with resistance to pyrethroids and organophosphates in salmon lice sampled from wild sea trout and Atlantic salmon in Norway in 2014. Multiresistant lice have the genotypes R-RR or R-RS, while fully sensitive lice are S-SS. Lice that are resistant to organophosphates only are S-RR or S-RS, while individuals that are resistant to pyrethroids only are R-SS.

registrations of multiresistant lice were thus from 2009 in Ireland (31%), Shetland (11%) and Northern Norway (7%). Already in the Scottish sample from 2002, there were no lice that were sensitive to organophosphates. In Canada, the development in organophosphate resistance was negative from 2002 to 2009, with an increasing frequency of the genotype RS, but the RR genotype was not detected until 2017. In the sample from Shetland in 2009, only one out of 96 lice tested was sensitive to pyrethroids.

## 3.3. Temporal and spatial development

When considering all sampling sites in the North Atlantic, the temporal cofactor is significant, (d.f. = 4, $\chi^2 = 211$, $p < 0.001$), with an average frequency of multiresistance of 0.0 in 2000, 0.03 in 2002, 0.1 in 2009, 0.5 in the 2014 samples and 0.2 in 2016. The general trend is an increase in the frequency of multiresistant lice with time. The drop observed between 2014 and 2016 (0.5 and 0.2) is explained by the large number of highly resistant samples from the west of Norway in 2014. If the samples from 2014 and 2016 are analysed together, the average frequency is 0.4. The sampling site was also a significant cofactor (d.f. = 5, $\chi^2 = 213$, $p < 0.001$), with an average multiresistance frequency of 0.01 in Canada, 0.02 in Sørlandet, 0.13 in Finnmark, 0.24 in Ireland, 0.34 in Scotland and 0.45 in Western Norway.

For the Norwegian samples, the sampling year was a significant cofactor (d.f. = 3, $\chi^2 = 232$, $p < 0.001$), with an estimated frequency of multiresistance of 0.0 in the 2000–2002 samples, 0.04 in the 2009 samples and 0.53 in the 2014 samples. The sampling site was also a significant cofactor (d.f. = 7, $\chi^2 = 109$, $p < 0.001$), with an estimated frequency of multiresistance of greater than 0.5 in the West Norway samples (Namsfjord, Trondheimsfjord, Romsdalsfjord, Storfjord, Sognefjord, Hardangerfjord and Rogaland), 0.19 in Finnmark and 0.02 in Sørlandet.

## 4. Discussion

We present the first estimate of the spaciotemporal evolution of genetic multiresistance in salmon lice in the North Atlantic. Resistant genotypes for organophosphates and pyrethroids were simultaneously found in lice from all parts of the North Atlantic, except from Canada. In regions with intensive aquaculture, salmon lice that are sensitive to both pyrethroids and organophosphates were almost extinct in the most recent samples. We, therefore, conclude that extensive chemical usage for delousing cage-reared farmed salmonids has led to and propagated multiresistant lice.

Throughout the North Atlantic, organophosphates were not used from the introduction of emamectin benzoate in 1999 until 2008, and pyrethroids were only used sparsely in the same period (tables 1 and 2). However, from 2008 and onwards, the use of both compounds increased dramatically in all regions where they were permitted. This coincides with reports of treatment failure after the use of emamectin benzoate [39], whereby fish farmers were forced to turn the delousing-toolbox upside down to find something that could control the lice once again. However, even though the pause in the use of organophosphates had resulted in temporarily increased levels of sensitivity [25], the resistant genotype was still present in lice in low or modest frequencies, and our most recent results indicate that the renewed selection has almost eradicated lice that were sensitive to organophosphates in many regions (figure 1). For pyrethroids, we have fewer intermediate data points, but the results from Norway demonstrate a similar negative development in the frequency of lice that are sensitive to pyrethroids. As a result, there are few lice sampled from the European side of the North Atlantic that are sensitive to both compounds, and the frequency of individuals that are resistant to both increased in the timeline of this study (figure 1).

The presence of multiresistant lice on wild Atlantic salmon demonstrates the mode of dispersal of resistance in a marine species. On land, pests either move by themselves, their hosts, and/or by human activity. In the sea, the currents and migrating hosts enable individual parasites to travel over vast distances and spread their genes, resistant or susceptible, to new areas. While the first reports of pyrethroid resistance were from Norwegian fish farms around 2000 [19], the highest frequency of pyrethroid resistance in the historical material is from Shetland in 2009. The treatment history of the site of origin is not known, thus it is possible that the high level of resistance is caused by a recent treatment. However, a laboratory strain originating from Shetland in 2011 also demonstrated a very high level of resistance, indicating that pyrethroid resistance was widespread [40]. Local use of pyrethroids on Shetland was highest in 2009 and 2010 (table 2). In Ireland, organophosphates were no longer available as a delousing chemical from 2013 and onwards (table 1). Still, the percentage of lice carrying either the RS or the RR genotype increased from 51.6% in 2009 to 80.3% in 2016. This continued increase in the frequency of resistance can be explained by the dispersal of lice from other aquaculture regions, such as Norway and Scotland. Notable is also the finding of a multiresistant louse in the Icelandic material, as neither of the compounds have been applied on Iceland.

The results of the present study indicate a limited, or perhaps no cost associated with being resistant to both organophosphates and pyrethroids simultaneously. High frequencies of multiresistant salmon lice sampled from farmed salmonids could be explained by recent treatments that have removed the sensitive lice. However, similarly high frequencies of multiresistance in lice sampled from wild salmonids in multiple regions cannot easily be explained with recent treatment incidents. In salmon lice sampled from wild sea trout in Norway, the frequency of lice resistant to both pyrethroids and organophosphates was almost 75% of the genotyped lice in one region, and more than 50% of the sampled lice in all regions with intensive aquaculture. It is highly unlikely that these lice were survivors from local delousing incidents that have changed host from farmed to wild salmonids. Previously, wild sea trout in Norway, which mainly remain within one fjord system, have been found to mirror the resistance frequencies of organophosphates and pyrethroids found in fish farms within the same region [24,26]. The most likely explanation is that sea trout and farmed salmonids share the same lice, and that intense aquaculture activity represents the primary driver of the evolution of this parasite in some regions [26]. This is also supported by observed aquaculture-driven changes in virulence in lice sampled from aquaculture dense regions [41].

As in land-based food production, multi-pesticide resistant parasites represent a threat to salmonid fish farming. In the most aquaculture-intense regions of Europe, and particularly in Norway and Scotland, individuals resistant to both organophosphates and pyrethroids are already widespread. On the American side of the North Atlantic, multiresistance is not yet documented, but it is likely that an introduction of pyrethroids in this region will drive multiresistance also here. In the Pacific, the wild hosts of *L. salmonis* (North America) and *C. rogercresseyi* (Chile) far outnumber the farmed salmonid population. The selection pressure from aquaculture on lice is thus less compared to Europe, where the farmed salmonids constitute the majority of salmon lice hosts [42]. However, in the intensely farmed regions of Chile, loss of sensitivity to both organophosphates and pyrethroids have been documented in *C. rogercresseyi* [43,44]. While the genetic markers of resistance to both compounds are not yet in place in this species, it is reasonable to assume that multiresistant individuals already exist also in this region or will appear within the near future. Furthermore, this study has only investigated simultaneous resistance to pyrethroids and organophosphates in the North Atlantic, but it is also possible that there are individuals that are resistant to emamectin benzoate and/or hydrogen peroxide in combination with pyrethroids and/or organophosphates.

Ethics. The salmon louse is not protected by Norwegian legislation or by the European Convention for the protection of Vertebrate Animals used for Experimental and other Scientific Purposes, but the host is. In the cases where lice were sampled from farmed fish, this was done with the consent of the owner and in connection with routine louse counting. When lice were sampled from wild salmonids, this was done in cooperation with ongoing research projects that had a licence to capture wild salmonids, or by local fishermen that had a licence to harvest wild salmonids. The study did not lead to extra mortality on wild salmonid populations.

Data accessibility. A dataset supporting this article is provided in the electronic supplementary material [45].

Authors' contributions. H.B.F., A.S., F.N., P.A.B., A.-K.T. and K.A.G. conceived and designed the study. H.B.F. and K.A.G. coordinated the work. H.B.F. and F.B. conducted statistical analyses, while V.T.A. conducted genotyping. H.B.F. wrote the first draft of the manuscript together with K.A.G. All authors contributed to data interpretation and the writing of the final manuscript.

Competing interests. We declare we have no competing interests.

Funding. This study was financed by resources from the Norwegian Ministry of Trade and Industry (NFD), the Norwegian Food Safety Authority (Mattilsynet) and Regional Research Fund (RRF) grant no. 245912.

Acknowledgements. We would like to extend our gratitude to all of those who have contributed with lice samples from fish farms or wild salmonids from throughout the North Atlantic.

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
