## [Peer Review File · Royal Society Open Science]

Review History

RSOS-210265.R0 (Original submission)

Review form: Reviewer 1

Is the manuscript scientifically sound in its present form?

Yes

Are the interpretations and conclusions justified by the results?

Yes

Is the language acceptable?

Yes

Do you have any ethical concerns with this paper?

No

Have you any concerns about statistical analyses in this paper?

No

Recommendation?

Accept with minor revision (please list in comments)

Comments to the Author(s)

This is very nicely written concise report on temporal and spatial distribution of multiresistance in salmon lice for two commonly used pesticides, organophosphate and pyrethroid across North Atlantic region. The authors have employed a novel genetic method for mapping simultaneously two genetic markers related to or directly affecting pesticide resistance in salmon lice.

Salmon lice have become one of the most serious hazards for salmon farming, and the reported results do not only reveal the potential threat of losing important tools in fight against these pests, but also lays out the speed of evolution imposed by intensive farming on parasites and thus makes an important contribution to the field.

I have a few minor comments that might help readers to understand some of the presented points without need to check further literature.

Introduction L59 onward. Make it clear already here that this part considers the history of the use of the pesticides and that the data in the ms considers the time when they were re-introduced from 2000 onwards. Reading this text and tables 1&2 gives first an impression of contradiction, as here it is told that organophosphates were first pesticides used, while tables 1&2 on historical use tell the opposite. Explanation for this comes only in second chapter of discussion.

A short explanation of the reproduction biology of salmon louse and its dispersal would help to understand how pesticide resistance can spread fast among populations.

L115 What is PatoGen AS?

L152 Can you provide data on intensity of aquaculture in Norway in different regions, e.g. in supplementary file?

Fig 1& 2. Order and colors of the coding are not intuitive. It is very hard to figure out the proportion of different genotypes. It would be easier if both genotypes were in the same order, now pyrethroid resistance is in order sensitive-resistant, while organophosphate resistance is in opposite order. Also, if the text explaining the figure contents was in the same order than the figure legend, it would help reading the figure. The color scheme red-green is also not friendly to persons with impaired color-vision and should be considered revising.

Decision letter (RSOS-210265.R0)

Dear Mrs Fjørtoft

On behalf of the Editors, we are pleased to inform you that your Manuscript RSOS-210265 "Losing the "arms race": Multiresistant salmon lice are dispersed throughout the North Atlantic Ocean" has been accepted for publication in Royal Society Open Science subject to minor revision in accordance with the referees' reports. Please find the referees' comments along with any feedback from the Editors below my signature.

Please submit your revised manuscript and required files (see below) no later than 7 days from today's (ie 12-Apr-2021) date. Note: the ScholarOne system will 'lock' if submission of the revision is attempted 7 or more days after the deadline. If you do not think you will be able to meet this deadline please contact the editorial office immediately.

on behalf of Dr Punidan Jeyasingh (Associate Editor) and Kevin Padian (Subject Editor)
openscience@royalsociety.org

Associate Editor Comments to Author (Dr Punidan Jeyasingh):

Comments to the Author:

My apologies to the authors. I have had a hard time securing reviews for this manuscript. I did secure an expert review, which was quite positive and constructive. With much gratitude to the expert, I invite the authors to make these minor revisions and resubmit a fresh version.

Subject Editor Comments to Author (Professor Kevin Padian):

Comments to the Author:

Thanks for your submission. We are happy to accept it pending your attention to the comments of our reviewer. Best wishes.

Reviewer comments to Author:

Reviewer: 1

Comments to the Author(s)

This is very nicely written concise report on temporal and spatial distribution of multiresistance in salmon lice for two commonly used pesticides, organophosphate and pyrethroid across North Atlantic region. The authors have employed a novel genetic method for mapping simultaneously two genetic markers related to or directly affecting pesticide resistance in salmon lice.

Salmon lice have become one of the most serious hazards for salmon farming, and the reported results do not only reveal the potential threat of losing important tools in fight against these pests, but also lays out the speed of evolution imposed by intensive farming on parasites and thus makes an important contribution to the field.

I have a few minor comments that might help readers to understand some of the presented points without need to check further literature.

Introduction L59 onward. Make it clear already here that this part considers the history of the use of the pesticides and that the data in the ms considers the time when they were re-introduced from 2000 onwards. Reading this text and tables 1&2 gives first an impression of contradiction, as here it is told that organophosphates were first pesticides used, while tables 1&2 on historical use tell the opposite. Explanation for this comes only in second chapter of discussion.

A short explanation of the reproduction biology of salmon louse and its dispersal would help to understand how pesticide resistance can spread fast among populations.

L115 What is PatoGen AS?

L152 Can you provide data on intensity of aquaculture in Norway in different regions, e.g. in supplementary file?

Fig 1& 2. Order and colors of the coding are not intuitive. It is very hard to figure out the proportion of different genotypes. It would be easier if both genotypes were in the same order, now pyrethroid resistance is in order sensitive-resistant, while organophosphate resistance is in opposite order. Also, if the text explaining the figure contents was in the same order than the figure legend, it would help reading the figure. The color scheme red-green is also not friendly to persons with impaired color-vision and should be considered revising.

===PREPARING YOUR MANUSCRIPT===

===PREPARING YOUR REVISION IN SCHOLARONE===

Author's Response to Decision Letter for (RSOS-210265.R0)

See Appendix A.

Decision letter (RSOS-210265.R1)

Dear Mrs Fjørtoft,

It is a pleasure to accept your manuscript entitled "Losing the "arms race": Multiresistant salmon lice are dispersed throughout the North Atlantic Ocean" in its current form for publication in Royal Society Open Science. The comments of the reviewer(s) who reviewed your manuscript are included at the foot of this letter.

on behalf of Dr Punidan Jeyasingh (Associate Editor) and Kevin Padian (Subject Editor)
openscience@royalsociety.org

Associate Editor Comments to Author (Dr Punidan Jeyasingh):

I thank the authors for incorporating reviewer comments. I am happy to recommend this version for publication.

Appendix A

Response to reviewer

Thank you for your encouraging and constructive response. We have tried to improve the manuscript as suggested.

Reviewer comment:

Introduction L59 onward. Make it clear already here that this part considers the history of the use of the pesticides and that the data in the ms considers the time when they were re-introduced from 2000 onwards. Reading this text and tables 1&2 gives first an impression of contradiction, as here it is told that organophosphates were first pesticides used, while tables 1&2 on historical use tell the opposite. Explanation for this comes only in second chapter of discussion.

Response:

We see that this could be confusing and have tried to make this clearer in the text.

Reviewer comment:

A short explanation of the reproduction biology of salmon louse and its dispersal would help to understand how pesticide resistance can spread fast among populations.

Response: A short section on the life cycle and modes of dispersal has been included.

Reviewer comment:

L115 What is PatoGen AS?

Response:

We have explained this better in the revised text.

Reviewer comment:

L152 Can you provide data on intensity of aquaculture in Norway in different regions, e.g. in supplementary file?

Response:

A figure illustrating yearly production in the different regions of Norway has been included in the supplementary material.

Reviewer comment:

Fig 1& 2. Order and colors of the coding are not intuitive. It is very hard to figure out the proportion of different genotypes. It would be easier if both genotypes were in the same order, now pyrethroid

resistance is in order sensitive-resistant, while organophosphate resistance is in opposite order. Also, if the text explaining the figure contents was in the same order than the figure legend, it would help reading the figure. The color scheme red-green is also not friendly to persons with impaired color-vision and should be considered revising.

Response:

We have changed the order of the genotypes so that it hopefully is more intuitive. We have also changed the color scheme